# The Endocrine-Disrupting Chemical Benzophenone-3 in Concentrations Ranging from 0.001 to 10 µM Does Not Affect the Human Decidualization Process in an In Vitro Setting

**DOI:** 10.3390/ijms26199314

**Published:** 2025-09-24

**Authors:** Kristin Krausser, Julia Howanski, Beate Fink, Mario Bauer, Florence Fischer, Federica Romanelli, Ana Claudia Zenclussen, Anne Schumacher

**Affiliations:** 1Department of Environmental Immunology, Helmholtz-Centre for Environmental Research (UFZ), 04318 Leipzig, Germany; kristin.krausser@ufz.de (K.K.); julia.howanski@ufz.de (J.H.); beate.fink@ufz.de (B.F.); mario.bauer@ufz.de (M.B.); florence.fischer@ufz.de (F.F.); federica.romanelli@ufz.de (F.R.); ana.zenclussen@ufz.de (A.C.Z.); 2Saxon Incubator for Clinical Translation (SIKT), Perinatal Immunology, University Leipzig, 04103 Leipzig, Germany; 3Institute of Clinical Immunology, University Leipzig, 04109 Leipzig, Germany

**Keywords:** benzophenone-3, endocrine-disrupting chemicals, decidualization, embryo implantation, trophoblast attachment and invasion

## Abstract

Endocrine-disrupting chemicals such as benzophenone-3 (BP-3) can have severe consequences for human reproduction by affecting critical processes during pregnancy. To shed further light on potential harmful BP-3 actions, our current study addressed the impact of BP-3 on decidualization and trophoblast invasion. Decidualization was initiated in human endometrial stromal cells (THESC) upon treatment with a mixture of cAMP, progesterone, and estradiol. In parallel to hormonal treatment, the cells were exposed to different BP-3 concentrations ranging from 0.001 µM to 10 µM. The expression of decidualization and invasion markers was determined. Moreover, trophoblastic spheroids derived from JEG-3 cells were transferred to decidualized THESC after BP-3 exposure, and spheroid attachment and invasion were analyzed. Hormonal treatment successfully initiated decidualization in THESC, which was confirmed by increased prolactin levels and *IGFBP1* and *NCOA*-3 mRNA expression. Notably, BP-3 exposure did not affect these markers. Furthermore, BP-3 changed neither THESC proliferation nor viability nor the frequency of cells expressing MMP2/9 or TIMP1/3. Trophoblastic spheroid attachment and outgrowth into THESC were not altered through any of the BP-3 concentrations applied. Our results do not provide evidence for an influence of BP-3 on the decidualization process and the capability of trophoblast cells to adhere and invade into endometrial stromal cells.

## 1. Introduction

The increasing use of sunscreens to prevent skin cancer and skin ageing is accompanied by potential health risks associated with certain ingredients in these products [1]. In particular, benzophenone-3 (BP-3, also known as oxybenzone), a widely used ultraviolet (UV) filter, is suspected of interfering with hormonal processes in the human body as an endocrine disruptor (EDC) [2]. EDCs, due to their structural similarity to endogenous hormones, can modulate endocrine signaling pathways and thus disrupt physiological processes in the body [3]. Hormone-dependent processes such as the female menstrual cycle and the decidualization of the uterine lining, an essential step for successful embryo implantation and pregnancy, are particularly affected [4]. More precisely, decidualization is the transformation of stromal cells of the endometrium into a specialized cell population that creates an optimal environment for the implantation of blastocysts [5]. This cyclical transformation occurs independently of fertilization [6]. Studies suggest that the disruption of this process is associated with implantation failure, miscarriage, and pregnancy complications [7,8].

After a whole-body dermal application of 2 mg of sunscreen at a concentration of 10% (wt/wt) per cm^2^ body surface in women and men, BP-3 is absorbed through the skin and can be detected in its parent form in plasma and urine. For women, maximum plasma levels of approximately 200 ng/ mL BP-3 were reached 3–4 h after application. For men, maximum plasma concentrations of 300 ng/mL BP-3 were reached 3 h after application. At 24 and 96 h after application of the active formulation, BP-3 plasma levels were not significantly different in either the women or the men, suggesting that BP-3 did not further accumulate [9]. Several other studies proved that, after skin penetration, BP-3 is present in blood (up to 238 ng/mL; equals 1.04 µM) [9,10], urine (up to 603 ng/mL, equals 2.64 µM) [9,11], amniotic fluid (up to 11.6 ng/mL, equals 0.05 µM) [12], the serum of pregnant women (up to 71.8 ng/mL, equals 0.32 µM) [13], and even breast milk (up to 8.20 ng/mL, equals 0.04 µM) [14]. Furthermore, BP-3 can be metabolized in the human liver by phase I and phase II biotransformations mediated via the human cytochrome P450 enzymes [15].

Thus, its influence on decidualization is considered very likely. Nevertheless, there are hardly any studies to date that have investigated the effect of BP-3 on decidualization and the subsequent implantation of the embryo.

Various molecules make a decisive contribution to the decidualization process and can be used as markers for successful decidualization. These include prolactin (PRL), the insulin-like growth factor binding protein 1 (IGFBP1) and metalloproteinases (MMPs), and their inhibitors (TIMPs). PRL is strongly induced during decidualization, and large amounts of PRL are released by decidual cells as the differentiation process progresses [16], while IGFBP1 plays a central role in cell migration and adhesion [17]. MMP2 and MMP9 catalyze matrix degradation [18,19], whereas TIMP1 and 3 regulate these processes and prevent excessive tissue degradation [20,21]. Moreover, nuclear receptor coactivator 3 (NCOA3) acts as a cofactor for progesterone action and influences gene expression during decidualization [22,23].

In the present study, we first investigated whether BP-3 concentrations reflecting human exposure conditions [9] affect viability, proliferation, and the expression of decidualization-associated markers in the human endometrial stromal cell line, THESC, after initiation of decidualization. Although THESC cells represent a telomerase-transformed cell line, they are frequently used in reproductive medicine and are recognized by the research community as a suitable model for studying decidualization [24,25] and factors affecting this process, including EDCs [26,27]. The reason for using endometrial stromal cell lines instead of primary endometrial stromal cells is often the lack of continuous availability of endometrial tissue obtained from patients after hysterectomy. Furthermore, Saleh et al. concluded that stimulants such as those used in the present study to initiate decidualization produced comparable results in THESC cells and primary endometrial stromal cells [28].

In a second step, we used our established heterologous in vitro implantation model to investigate a potential impact of BP-3 on the ability of trophoblastic spheroids derived from JEG-3 cells to adhere to and invade into decidualized endometrial stromal cells [29].

## 2. Results

### 2.1. BP-3 Did Not Modulate mRNA and Protein Expression of Decidualization Markers in THESC

First, we confirmed the initiation of decidualization in THESC following hormone treatment via an increase in characteristic decidualization markers at the mRNA level (*PRL*, *IGFBP1*, *NCOA3*) and at the protein level (PRL), by comparing cell cultures with and without hormones (Figure 1A–D). Then, we were interested in evaluating whether BP-3 treatment throughout decidualization would affect this process. Our results showed that none of the BP-3 concentrations applied had an impact on THESC decidualization. Comparable expression levels of *PRL*, *IGFBP1*, and *NCOA3* were detected between different BP-3 groups and the respective control group (hormones plus ethanol) (Figure 1A–D).

### 2.2. BP-3 Did Not Affect Viability, Proliferation, and Expression of Metalloproteinases and Their Inhibitors in THESC

Next, we wondered whether hormonal treatment alone or the presence of different BP-3 concentrations could affect general cellular parameters such as cell viability and proliferation following the induction of THESC decidualization. Hormonal treatment did not provoke significant changes in THESC viability and proliferation (Figure 2A,B), although a decrease in the percentage of viable cells was visible after the addition of hormones (Figure 2A). BP-3 in various concentrations did not alter cell viability and proliferation when compared to the hormones plus ethanol control group (Figure 2A,B). Furthermore, we did not observe significant changes in the percentages of THESC expressing the pro-invasion markers MMP2, MMP9 or the anti-invasion markers TIMP1 and TIMP3 following hormonal treatment alone or together with BP-3 (Figure 2C–F). The gating strategy and representative histograms, including fluorescence minus one (FMO) controls of all molecules, are shown in Appendix A.

In-line analyses of intracellular expression levels (MFI) of the two metalloproteinases and their inhibitors did not reveal any significant alterations among the experimental groups and the respective control group (Figure 3A–D).

### 2.3. BP-3 Did Not Interfere with the Ability of Decidualized THESC to Allow Trophoblastic Spheroid Attachment and Invasion

Although our previous results did not suggest negative effects of BP-3 on THESC survival, proliferation, or the capability of these cells to decidualize, we sought to determine whether BP-3 influences THESC decidualization in such a way that it subsequently affects the attachment and invasion of trophoblast cells into a decidualized cell layer. We generated JEG-3 trophoblastic spheroids and confronted the spheroids with THESC previously decidualized in the presence of BP-3, mimicking the physiological process of embryo implantation into the maternal endometrium in an in vitro model. Afterwards, we determined the attachment rate and the outgrowth of trophoblastic spheroids, respectively, through the invasion into the THESC layer. Figure 4A shows representative pictures displaying a trophoblastic spheroid directly after transfer (0 h) to THESC, as well as an expanded/invaded spheroid after 24 h and 48 h. No statistically significant differences were found with respect to trophoblastic spheroid attachment to or outgrowth in decidualized THESC previously treated with BP-3 compared to the control group (Figure 4B).

## 3. Discussion

BP-3 is a widely used organic UV filter that has been detected in numerous human biological fluids [9,10,11,12], reflecting its ubiquitous environmental presence. However, the potential effects of BP-3 on critical reproductive processes, such as endometrial decidualization and embryo implantation, remain poorly understood. Therefore, this study aimed to investigate the impact of BP-3 exposure on endometrial stromal cells during decidualization and on subsequent implantation-related events using in vitro models. BP-3 concentrations between 0.001 and 10 µM were selected based on human biomonitoring data [9,30] and following the dose selection rationale described by Zhan et al. [31], who also based their BP-3 concentrations on levels detected in human biological samples. More precisely, Zhan et al. reviewed several studies presenting BP-3 concentrations in blood and urine ranging from non-detectable levels up to 95 µM, with more recent biomonitoring data indicating levels closer to 1 µM [31]. In addition, another study by Janjua et al. demonstrated that whole-body sunscreen application results in plasma concentrations of approximately 200–300 ng/mL (corresponding to 0.9–1.3 µM) [9]. Similar systemic absorption was confirmed in a randomized clinical trial by Matta et al., who showed that repeated sunscreen application led to detectable BP-3 levels in plasma above 0.5 ng/mL (≈0.002 µM) [30]. Accordingly, to mimic real BP-3 exposure conditions in the human body, we chose a concentration range reflecting both commonly reported exposure levels in the general population and higher concentrations measured after intensive sunscreen use.

In the present study, we first assessed the effects of BP-3 exposure on the viability and proliferation of human endometrial stromal cells (THESC) under decidualization conditions. The treatment of THESC with BP-3 at different concentrations did not result in any significant changes in cell viability. The study conducted by Zhan et al., which examined the effects of BP-1, BP-3, BP-8, and the metabolites M1 and M2 on primary human endometrial stromal cells (HESC), demonstrated a significant increase in apoptotic cells, with the strongest effect observed in the BP-3 group at concentrations between 0.01 and 0.1 µM [31]. This induction of apoptosis, assessed through Annexin V staining and flow cytometry, was consistent with their finding that BP-3 at a concentration of 1 μM also significantly reduced cell viability, measured in a CCK-8 assay. Following the recommendations of Zhan et al., we analyzed cell proliferation using Ki67 staining but found no significant effect of BP-3 on our cell line. Furthermore, in contrast to their findings, our analyses of cell viability using a fixable viability dye (FVD) revealed no significant alterations. In addition to investigating apoptosis, Zhan et al. performed cell-cycle analysis using propidium iodide staining and flow cytometry, which revealed a BP-3-induced arrest in the G2/M phase in HESC cells. They attributed both the increase in apoptosis and the G2/M arrest to the upregulation of FOXO1. The discrepancy in cell viability may be attributed to the usage of primary versus immortalized stromal cells, as THESC typically exhibit reduced responsiveness to hormonal and stress stimuli and greater resistance to apoptosis compared to primary cells [24,28]. However, our observations are consistent with the results of Kerdivel et al., who found no proliferative effects of BP-3 in estrogen receptor α-positive MCF-7 breast cancer cells. Neither cell number nor DNA synthesis was significantly affected by BP-3 at concentrations of 0.01, 0.1, and 1 µM [32]. In contrast, LaPlante et al. reported a significant increase in Ki67 expression in the mammary gland epithelium of animals after perinatal exposure to the highest tested dose of BP-3 (3000 µg/kg/day ≈ 13.14 µM) in a mouse model. At lower doses (0.13 and 0.93 µM), no significant effect was observed [33]. Matouskova et al. found a significant decrease in Ki67-positive cells in the prepubertal mammary gland in female mice after low perinatal oxybenzone exposure at 30 µg/kg/day [34], further demonstrating that BP-3 can exert both pro- and antiproliferative effects, depending on dose, time window, target tissue, and in vitro/in vivo model used. The BP-3 effect on proliferative processes and cell viability seems to be cell type-specific and context-dependent, which emphasizes the need for further investigations—especially with regard to long-term or combined hormonal influences.

An increase in PRL secretion and increased mRNA expression of PRL and IGFBP1 confirmed the successful decidualization of THESC after hormone treatment, as also suggested by other research groups like Maurya et al. [22]. All groups of BP-3 concentrations tested showed comparable expression levels, so that an influence of BP-3 on decidualization was not detectable. Our results are consistent with the study by Skorkowska et al. [35], who investigated the effect of combined prenatal and adult dermal BP-3 exposure on PRL levels in the blood of female rats. They were also unable to observe any effect of BP-3 on PRL secretion. In contrast to our results, Zhan et al. [31] reported a significant upregulation of the decidualization markers PRL and IGFPB1, as well as LEFTY (which was not included in our analyses), following co-exposure to BP-3 (0.01 µM) and hormonal stimulation in HESC. Furthermore, they observed elevated expression levels of the transcription factors FOXO1 and STAT3, along with the hormone receptors estrogen receptor α and progesterone receptor. Given their recommendation to investigate additional nuclear receptors, our analysis of NCOA3 expression may complement and extend their findings. Although the central role of NCOA3 in mediating decidualization has been intensively studied, no comparable studies on the influence of BP-3 on this coactivator are available to date. Our data showed no effect of BP-3 on NCOA3 mRNA expression. However, different effects on NCOA3 have already been described for other endocrine disruptors such as bisphenol A (BPA). For instance, Salian et al. [36] showed increased NCOA3 protein expression in the testes of rats after neonatal BPA exposure. Overall, our findings indicate that BP-3 significantly affected neither NCOA3 expression nor the decidualization process at the genetic level, based on the set of decidualization markers examined in our study.

Finally, we were interested in investigating whether BP-3 would have an effect on the implantation process. Therefore, we first determined the expression of pro- and anti-invasive markers. Neither hormone treatment alone nor in combination with all tested BP-3 concentrations showed a significant influence on the expression of the pro-invasive markers MMP2 and MMP9 or the anti-invasive markers TIMP1 and TIMP3 in THESC. These data largely correspond to those of Hisamatsu et al., who, with the exception of TIMP3, also found no change in the expression of the MMPs and TIMPs we investigated during the decidualization of primary endometrial cells [37]. However, they contrast with previous work involving other endocrine disruptors such as BPA, which shares similar endocrine-modulating properties. Wang et al. [38] showed, in human trophoblastic BeWo cells, that even low concentrations of BPA (from 0.01 µM) led to a shift in the MMP/TIMP ratio. In line with our results, the study by Dominguez et al. [39] showed no significant effect of BPA on MMP9 activity, although a concentration-dependent increase in the total amount of MMP9 was observed. However, this only reached statistical significance at concentrations above 1 µg/mL—i.e., above physiologically relevant values [39]. Several other studies have shown that EDCs such as dioxin [40], hexachlorobenzene [41], and TCDD [42] can modulate the expression and activity of MMPs and TIMPs in endometrial stromal cells. In contrast, the effects of BP-3 have so far been investigated exclusively in other cell systems, with inconsistent results [43,44,45]. Our current experiments close this research gap by demonstrating that human-exposure-relevant concentrations of BP-3 do not significantly affect the expression of the investigated MMPs and TIMPs in THESC cells during decidualization.

Next to the investigation of the gene expression of anti- and pro-invasive molecules in the human trophoblast cell line Swan 71, the study by Abud et al. also assessed visible functional effects on cell migration using both a wound healing assay and a co-culture model of murine blastocysts with uterine epithelial cells (UECs). Interestingly, and in contrast with our findings, Abud et al. reported both a significantly reduced migratory capacity in the wound-healing assay and, in the co-culture model, a significantly decreased attachment rate after 60 h at the lowest (2 µg/L) and highest (200 µg/L) BP-3 concentrations, as well as a significantly smaller implantation area after 144 h at all tested concentrations [44]. In our in vitro implantation model using JEG-3 trophoblastic spheroids, no significant differences in attachment rates or outgrowth areas between treatment groups were observed within the first 48 h; nonetheless, later-stage effects after 48 h cannot be excluded. Our results are, however, consistent with the observations of Santamaría et al. [46], who also found no significant effects on implantation number and implantation area after dermal BP-3 exposure in pregnant C57BL/6J mice during early pregnancy. The different results provided through the study from Abud et al. may be explained by differences in experimental design. Abud et al. exposed both the murine endometrium and the blastocysts continuously to BP-3 throughout the implantation process [44]. In our in vitro model, only the endometrial compartment was exposed to BP-3 prior to co-culture with the trophoblastic spheroids, as our aim was to investigate the isolated effect of BP-3 on decidualization and its influence on subsequent trophoblast attachment. The effects described by Abud et al. may indicate a stronger influence of BP-3 on the blastocyst itself or on the dynamic embryo-endometrium interaction in an ex vivo setting [44]. Particularly, the reduced attachment rates observed in the blastocyst/UECs co-culture model at both the lowest and highest BP-3 concentrations reflect a typical non-monotonic dose-response pattern, which is a hallmark of many EDCs. The critical interplay between prior successful decidualization and a healthy, developmentally competent blastocyst should not be underestimated. Future studies should, therefore, specifically address the combined effects of BP-3 on both compartments—the endometrium and the embryo—to better understand potential additive or synergistic mechanisms of action.

## 4. Materials and Methods

### 4.1. Cell Lines

Human endometrial stromal cells (THESC) and JEG-3 choriocarcinoma cells were purchased from the American Type Culture Collection (ATCC, Manassas, VA, USA). The THESC were maintained in phenol red-free DMEM/F12 medium (Life Technologies Limited, Inchinnan, UK, containing 2.5 mM L-glutamine) supplemented with 10% charcoal-stripped fetal bovine serum (FBS, PAN Biotech GmbH, Aidenbach, Germany), 0.01% insulin (SigmaAldrich, Darmstadt, Germany), and 1% penicillin/streptomycin (P/S, Biowest, Nuaillé, France). JEG-3 cells were maintained in DMEM medium (Life Technologies Limited, Inchinnan, UK, containing 4 mM L-glutamine) supplemented with 10% FBS and 1% P/S. Both cell lines were cultured at 37 °C in a humidified atmosphere containing 5% CO_2_. At 80% confluency, cell lines were harvested for sub-culturing or used in in vitro assays.

### 4.2. Decidualization of THESC and BP-3 Exposure

To induce in vitro decidualization, THESC were seeded at 1.2 × 10^5^ cells per well in 24-well plates and cultured in the aforementioned medium. Then, 24 h after seeding, cells were treated with 10 nM of estrogen (E2, 17β-estradiol; Merck, Darmstadt, Germany), 0.5 mM of 8-bromoadenosine 3′,5′-cyclic adenosine monophosphate (cAMP; Merck, Darmstadt, Germany), and 1 μM of medroxyprogesteron acetat (6α-Methyl-17α-hydroxyprogesteron-acetat) (MPA; Merck, Darmstadt, Germany) for 6 d to induce the decidualization process. E2 was prepared as a 10 mM stock solution in ethanol and serially diluted in medium to reach a final concentration of 10 nM. cAMP was dissolved in PBS to obtain a stock solution of 5 mM and diluted in medium to a final concentration of 0.5 mM. MPA was prepared as 10 mM stock solution in ethanol and diluted in medium for a final concentration of 1 µM. All stock solutions were stored at −20 °C until use. In parallel, some cellular approaches were exposed to various concentrations of BP-3 (Merck, Darmstadt, Germany) (0.001 to 10 µM dissolved in 0.01% ethanol; Chemsolute, Renningen, Germany). BP-3 was initially prepared in ethanol as 200 mM solution, diluted to 10 mM, aliquoted, and stored at −20 °C until use. After thawing, serial dilutions in medium were performed to generate the final concentrations of 0.001 to 10 µM. Cells cultured without hormones (named medium) or in the presence of 0.01% ethanol and hormones but without BP-3 (named hormones plus ethanol) served as controls. In all approaches, the culture medium containing the respective supplements (hormones, ethanol, and/or BP-3 or none) was refreshed every second day. After 6 d of culture, supernatants were collected and used for PRL protein analysis via an enzyme-linked immunosorbent assay (ELISA), and cell lysates were resuspended and stored in TRIzol™ Reagent (Life Technologies Corporation, MA, USA) at −80 °C for gene expression analyses.

### 4.3. Determination of PRL Secretion by ELISA

To measure PRL levels in cell supernatants, an ELISA was performed using the Invitrogen Prolactin ELISA Kit (Thermo Fisher Scientific, MA, USA) according to the manufacturer’s instructions. Briefly, 50 μL of standards and supernatants were added to a pre-coated plate, followed by incubation for 60 min at room temperature on an orbital shaker. Afterwards, the plate was washed four times with 300 μL of wash buffer. Then, 50 μL of PRL conjugate was added to each well and incubated for another 60 min. After another washing step, 100 μL of TMB substrate was added to each well, and the plate was incubated for 30 min at room temperature. The reaction was stopped by adding 50 μL of stop solution, and absorbance was measured at 450 nm with a correction at 570 nm using the Synergy H1 plate reader (BioTek, Shoreline, WA, USA).

### 4.4. Gene Expression Analyses of PRL, IGFBP1, and NCOA3

Total RNA was isolated using TRIzol^®^ reagent according to the manufacturer’s instructions. The extraction process included phase separation with chloroform (AppliChem GmbH, Darmstadt, Germany), RNA precipitation with isopropanol (Carl Roth, Karlsruhe, Germany) and glycogen (BIOgenetix, Darza, Romania), washing with 75% ethanol, and resuspension in RNase-free water (AppliChem GmbH, Darmstadt, Germany). RNA quantification was performed by measuring UV absorbance at 260 nm, and purity was assessed with the 260/280 nm ratio using a Nano Quant Infinite M200 (Tecan, Salzburg, Austria). The extracted RNA was stored at −80 °C until further use. For cDNA synthesis, a mix containing 0.8 µL of nuclease-free water (AppliChem, Darmstadt, Germany), 0.25 µL of random hexamer primer (Thermo Fisher, MA, USA), and 0.25 µL of oligo (dT)18 primer (Thermo Fisher, MA, USA) was prepared, added to RNA, incubated at 70 °C for 5 min in a thermocycler (Analytik Jena, Jena, Germany), chilled on ice for 5 min, and finally centrifuged. Next, a master mix was prepared for each sample by combining 4 µL of 5 × ImProm-II™ reaction buffer (Promega, Madison, WI, USA), 1 µL of 10 mM dNTP mix (ThermoFisher, MA, USA), 1.2 µL of 50 mM MgCl_2_ from the BIOTAQ™ DNA Polymerase kit (Bioline, London, UK), 1 µL of ImProm-II™ Reverse Transcriptase (Promega, Madison, WI, USA), and 0.5 of µL RiboLock RNase Inhibitor (40 U/µL, Thermo Fisher, MA, USA). After mixing, 7.7 µL of the master mix was added to each RNA sample. Reverse transcription was performed under the following conditions: 10 min at 25 °C, 60 min at 42 °C, and 10 min at 70 °C. The obtained cDNA was stored at −20 °C until further use.

### 4.5. Quantitative Real-Time PCR (qPCR)

Prior to the quantitative real-time PCR (qPCR), a pre-amplification step was performed to enhance the detection of target genes. For this, 1 µL of 1:10 diluted cDNA per sample was used in a reaction containing a Pre-Amp Mix. The Pre-Amp Mix consisted of 10× buffer (BioCat GmbH, Heidelberg, Germany), MgCl_2_ (BioCat GmbH, Heidelberg, Germany), a dNTP mix (Solis BioDyne, Tartu, Estonia), molecular biology-grade water (AppliChem, Germany), BioTaq polymerase (BioCat), and a 5× pooled primer mix. The primer mix was dissolved in low EDTA-TE buffer, consisting of Tris buffer, water, and TE buffer (AppliChem), and it included outer (out) forward and reverse primers for the target genes *IGFBP1* and *PRL*, as well as forward and reverse primers for the target gene, *NCOA3*, and the reference genes, *GAPDH* and *HPRT1* (Table 1). All primers were ordered from Sigma-Aldrich (Merck KGaA, Darmstadt, Germany). Pre-amplification was carried out in 12 cycles using the BioMetra TAdvanced thermocycler (Analytik Jena, Jena, Germany). Subsequently, a qPCR analysis was conducted using 2 µL of the 1:10 diluted pre-amplified samples mixed with a master mix. The master mix contained 10× buffer, molecular biology-grade water, MgCl_2_, a dNTP mix, BioTaq polymerase, and 20× EvaGreen dye (BIOMOL GmbH, Germany), along with specific inner (in) primer pairs for *IGFBP1* and *PRL,* as well as specific primer pairs for *NCOA3*, *GAPDH*, and *HPRT1*. Target genes were analyzed in quadruplicate, while reference genes were analyzed in duplicate. Sample preparation was performed using the CyBi-SELMA pipetting system (Analytik Jena, Jena, Germany). qPCR was conducted on a LightCycler 480 system (Roche Diagnostics, Risch-Rotkreuz, Switzerland) using 45 cycles, and fluorescence signals were measured using the SYBR Green filter. Relative gene expression was quantified by normalizing the expression levels of the genes of interest to the geometric mean of the two reference genes (*GAPDH* and *HPRT1*). This normalization was performed individually for each sample.

### 4.6. Flow Cytometry Analyses for MMP2, MMP9, TIMP1, and TIMP3

Flow cytometry (FC) analyses were performed to determine the frequencies of cells expressing MMP2, MMP9, TIMP1, and TIMP3, as well as the degree of intracellular expression levels of all these molecules in decidualized THESC after BP-3 exposure. After chemical exposure, decidualized THESC were washed with phosphate-buffered saline (PBS; Biowest, Nuaillé, France) and transferred to round-bottom polystyrene tubes. To prepare cells for extra- and intracellular staining, 1 mL of FC buffer (PBS + 1% FCS) was added to each tube, followed by a centrifugation step for 10 min at 300× *g* and 4 °C. For extracellular staining, cells were incubated in a 100 μL solution containing FVD, eFluor 506) diluted 1:400 in FC buffer for 30 min at 4 °C. After extracellular staining, cells were washed in 1 mL FC buffer, centrifuged at 300× *g* for 10 min and 4 °C, and incubated in 100 μL of Foxp3/Transcription Factor Staining Buffer (Life Technologies Corporation, MA, USA) for 30 min at 4 °C in the dark for fixation and permeabilization. After another washing step in FC buffer, cells were intracellularly stained in 100 μL of antibody solution containing AF 647 Ki67 (clone: B56; dilution: 1:200), PE MMP2 (clone: 1A10; dilution:1:200), FITC MMP9 (clone: 56129; dilution: 1:200), AF 405 TIMP1 (clone: 63515; dilution: 1:400), and AF 750 TIMP3 (clone: 277128; dilution: 1:200) in permeabilization buffer for 30 min at 4 °C in the dark. After a final washing step in permeabilization buffer, cells were resuspended in 500 μL FC buffer and read on an Attune Nxt Acoustic Focusing Cytometer (AFC2; Life Technologies Holding Pte Ltd., Marsiling Ind Estate Road 3, Singapore).

### 4.7. Confrontation Assay Between JEG-3 Trophoblastic Spheroids and Decidualized THESC

An in vitro heterologous implantation model was applied to mimic the physiological process of embryo implantation into the maternal endometrium. In this model, JEG-3 trophoblastic spheroids, as the embryonic surrogate, were confronted with decidualized THESC as the maternal endometrial surrogate. Trophoblastic spheroids were generated by seeding 4 × 10^5^ JEG-3 cells in 35 mm Petri dishes in 1.5 mL of the aforementioned JEG-3 medium and letting them rotate at 70 rpm between 40 h and 48 h on an orbital shaker. After 2 d, two spheroids of similar size were transferred into one well containing a confluent layer of THESC. THESC were decidualized as previously described in the presence or absence of BP-3. Throughout trophoblast attachment and invasion, the medium did not contain hormones or BP-3. Confrontation lasted for a maximum of 48 h. After 0 h, 24 h, and 48 h, the spheroid size was measured using a microscope equipped with a camera (Keyence Microscope BZ-X800, Kyoto, Japan). Trophoblastic outgrowth was determined by subtracting the initial spheroid size (0 h) from the size measured at 24 h or 48 h using ImageJ software version 1.51.

### 4.8. Statistics

The data were analyzed using GraphPad Prism Version 10. Normal distribution was tested with the Shapiro–Wilk test. Depending on normality, we performed either parametric or nonparametric tests, followed by multiple comparison tests, as indicated in the respective figure legends. *p*-values below 0.05 were considered statistically significant.

## 5. Conclusions and Limitations of the Study

In summary, our findings indicate that human exposure-relevant concentrations of BP-3 do not significantly affect the viability, proliferation, or decidualization capacity of THESC. Likewise, no alterations were observed in the expression of key implantation markers or in trophoblastic spheroid adhesion and invasion to decidualized cells after BP-3 exposure. These results suggest that BP-3 has only a limited effect on endometrial function in THESC under the conditions tested. Nevertheless, our results must be interpreted with caution, as they relate exclusively to an endometrial cell line. They cannot, therefore, reflect the phenotypic and functional characteristics of primary endometrial cells and trophoblast cells or the decidualization and embryo implantation process in a complex organism. Furthermore, it was beyond the scope of this study to investigate the effects of pure BP-3 without hormone stimulation on endometrial cells and trophoblast cells. These limitations in the experimental design of our study could influence the overall interpretation of a potential BP-3 effect on decidualization and trophoblast invasion. Therefore, discrepancies with previous studies using primary cells or in vivo models highlight the need for further investigation to assess potential reproductive risks, including effects on both maternal and blastocyst compartments.

## Figures and Tables

**Figure 1 ijms-26-09314-f001:**
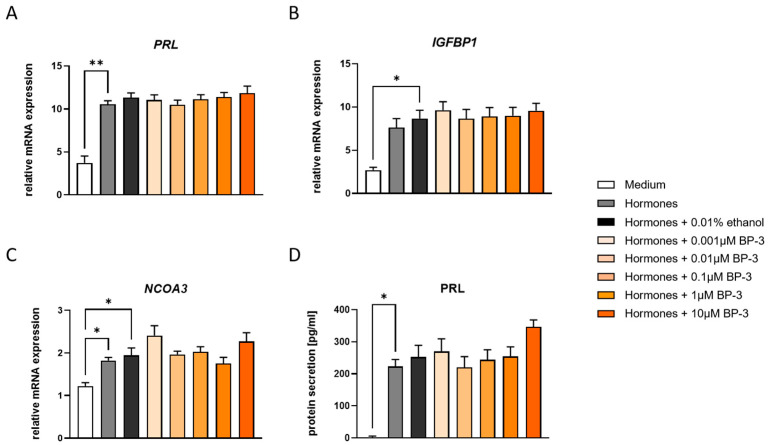
BP-3 did not affect decidualization markers in THESC. Relative mRNA expression of *PRL* (**A**), *IGFBP1* (**B**), and *NCOA3* (**C**), as well as PRL secretion (**D**), is displayed after THESC were decidualized in the presence or absence of BP-3. Data are shown as means plus standard deviation of means of three to four independent experiments. A one-way ANOVA, followed by Tukey’s multiple comparison test, was used to assess statistical differences. * *p* < 0.05; ** *p* < 0.01. *PRL*—prolactin, *IGFBP1*—insulin-like growth factor binding protein 1, *NCOA3*—nuclear receptor coactivator 3, BP-3—benzophenone-3.

**Figure 2 ijms-26-09314-f002:**
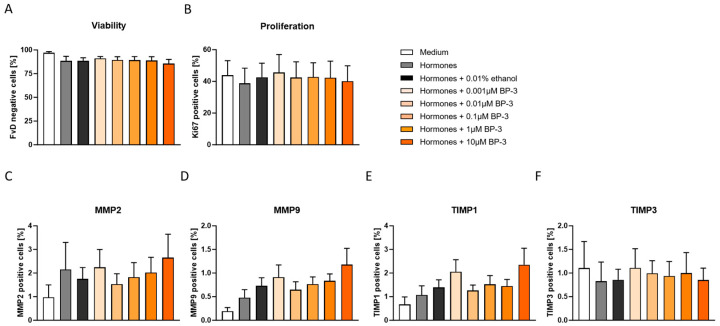
BP-3 did not affect THESC viability, proliferation, or the expression of pro- and anti-invasion markers. Viability (**A**), proliferation (**B**), and the percentage of THESC expressing the pro-invasive markers MMP2 (**C**) or MMP9 (**D**), as well as the non-invasive markers TIMP-1 (**E**) and TIMP-3 (**F**), are displayed after cells were decidualized in the presence or absence of BP-3. Data are shown as means plus standard deviations of means of four independent experiments. A one-way ANOVA followed by Tukey’s multiple comparison test was used to assess statistical differences. MMP—matrix metalloproteinase, TIMP—tissue inhibitor of metalloproteinase, BP-3—benzophenone-3.

**Figure 3 ijms-26-09314-f003:**
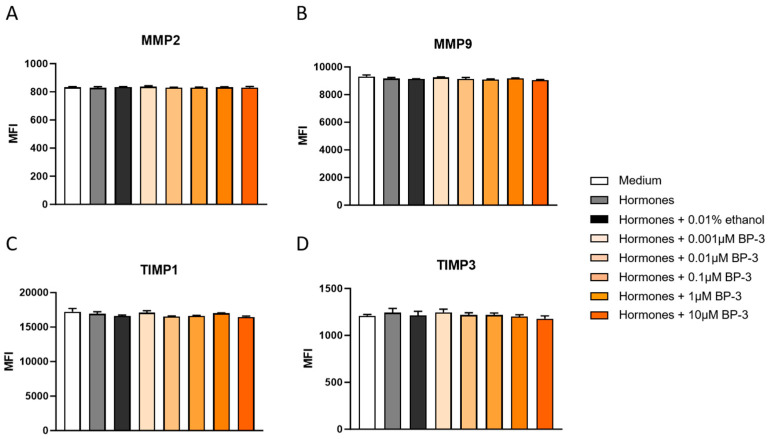
BP-3 did not affect the expression of pro- and anti-invasion markers in THESC. The intracellular expression of MMP2 (**A**), MMP9 (**B**), TIMP1 (**C**), and TIMP3 (**D**) is displayed after THESC were decidualized in the presence or absence of BP-3. Data are shown as means plus standard deviations of means of four independent experiments. A one-way ANOVA, followed by Tukey’s multiple comparison test, was used to assess statistical differences. MMP—matrix metalloproteinase, TIMP—tissue inhibitor of metalloproteinase, MFI—median fluorescence intensity, BP-3—benzophenone-3.

**Figure 4 ijms-26-09314-f004:**
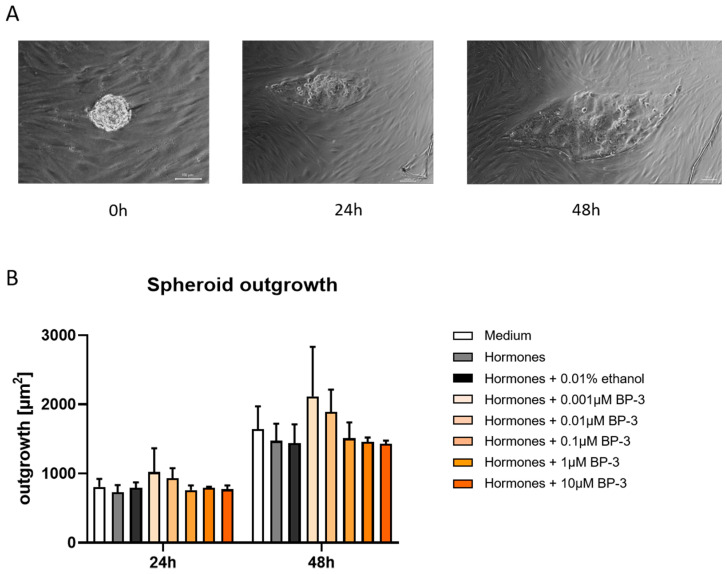
BP-3 did not affect the attachment or invasion of trophoblastic spheroids to decidualized THESC. Representative pictures of an attached and invading spheroid is shown at different time points (scale bar = 100 µM) (**A**), as well as spheroid outgrowth into THESC after cells have been decidualized in the presence or absence of BP-3 (**B**). Data are shown as means plus standard deviations of means of 3 independent experiments. A two-way ANOVA, followed by Sidak’s multiple comparison test, was used to assess statistical differences. BP-3—benzophenone-3.

**Table 1 ijms-26-09314-t001:** Overview of primers used in this study.

Gene	Nested PCR	Forward Primer	Reverse Primer
*IGFBP1*	in	TCCTTTGGGACGCCATCAGTAC	GATGTCTCCTGTGCCTTGGCTA
	out	GAGCACGGAGATAACTGAGGA	TGTTGCAGTTTGGCAGGTAA
*PRL*	in	GACCCTTCGAGACCTGTTTG	ATTCATCTGTTGGGCTTGCT
	out	GGTGTCAAACCTGCTCCTGT	TGCTGACTATCAGGCTCAGAAA
*NCOA3*		AGCTGAGCTGCGAGGAAA	GAGTCCACCATCCAGCAAGT
*GAPDH*		GAGTCAACGGATTTGGTCGT	TTGATTTTGGAGGGATCTCG
*HPRT1*		TGACACTGGCAAAACAATGCA	GGTCCTTTTCACCAGCAAGCT

## Data Availability

The raw data supporting the conclusions of this article will be made available by the authors upon request.

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
