# Peer review of "The Endocrine-Disrupting Chemical Benzophenone-3 in Concentrations Ranging from 0.001 to 10 µM Does Not Affect the Human Decidualization Process in an In Vitro Setting"

_ijms, 2025, doi:10.3390/ijms26199314_

Round 1

Reviewer 1 Report

Comments and Suggestions for Authors

Comments:

This study examines the effects of benzophenone-3 (BP-3) on human endometrial metaplasia and trophoblast invasion. It provides experimental evidence to elucidate the relationship between environmental factors and reproductive health. However, if the following limitations can be appropriately addressed, the manuscript holds considerable potential to make a meaningful contribution to the field of environmental reproductive health.

  1. Although the title accurately reflects the content of the manuscript, the experimental data only demonstrate that no significant effect was observed in the THESC model. Therefore, these findings may not be generalizable to humans, and the conclusions should be revised to avoid overly definitive statements. Additionally, specifying the dose range of BP-3 in the title would have enhanced the clarity and relevance of the study.
  2. The introductory section offers an oversimplified overview. By incorporating a discussion on the research significance of selecting the THESC model for this study, the introduction would more effectively emphasize the unique value derived from inter-model differences.
  3. The assertions presented in lines 51–52 and 152–153 of the article seem to be excessively definitive. A more nuanced approach to these descriptions would enhance their clarity and overall effectiveness.
  4. The absence of a dedicated BP-3 treatment group in Figures 1–3 represents a significant limitation in the experimental design. As a result, it has not been possible to clarify either the cytotoxic or the anti-proliferative effects of BP-3 on THESCs, nor to investigate any potential interactions between BP-3 and hormones during their decidualization process. Accordingly, the validity of the conclusion that "BP-3 does not affect decidualization" is compromised. It is therefore recommended that this control group be included in future experiments or that the conclusions drawn from the current study be strictly constrained in scope.
  5. In the Results section, Figures 2 and 3 do not indicate a difference in the expression of pro-invasion markers, MMP2 and MMP9, as well as anti-invasion markers, TIMP1 and TIMP3, between the Hormone group and the Medium group. This finding is inconsistent with the typical process of decidualization; however, the authors fail to provide an explanation for this discrepancy. Does this suggest that the decidualization model may not accurately replicate physiological decidualization?
  6. The co-culture model involving JEG-3 trophoblastic spheroids and decidualized THESC cells presents several methodological limitations. First, the functional experiments exclusively assessed trophoblast invasion without evaluating the direct effects of BP-3 on the trophoblasts themselves. Second, the co-culture medium was devoid of both hormones and BP-3. Notably, the absence of a positive control and insufficient justification for omitting hormones raise significant concerns. Third, the experiment was conducted over a limited 48-hour period, whereas the implantation window is critical and spans 72 hours. Therefore, this timeframe may have overlooked later-stage effects. It is recommended that the observation period be extended to 72 hours and that a positive control be included to validate the sensitivity of the model. If experimental supplementation is not feasible, it is imperative that any discussion rigorously analyzes how these limitations constrain the conclusions drawn from this study.

7. It is essential to address the limitations of the research. The constraints of the study should be further elucidated in light of the shortcomings in the experimental design and the inherent limitations of THESC, thereby enhancing the rigor of the literature and preventing overgeneralization of the conclusions.

Reviewer 2 Report

Comments and Suggestions for Authors

The study is described in an interesting way with many details. However, some important information is still missing:

Introduction:

There is no data on the fate of the substance in the body (data on BP-3 absorption through the skin, detectable levels in urine, serum, metabolism). Line 377:  “…our findings indicate that physiologically relevant concentrations of BP-...” - physiologically relevant?? Please provide information on how the test concentrations of BP-3 were selected.

Methods:

Methods are described in details (even sometimes too many details) but despite this I cannot find information on:
- how each test chemical (hormones and BP-3) was prepared (solvent, stock concentration, storage)

- No information about L-Glutamine content in both media

It is also a pity that the study design did not include the examination of some mechanisms of cytotoxicity, at least the cell cycle.

The presentation of the results would be more interesting if original graphs/histograms were included, e.g. from flow cytometry.
